# The N6-methyladenosine demethylase ALKBH5 regulates the hypoxic HBV transcriptome

Senko Tsukuda[1]*, James M. Harris[1], Andrea Magri[1], Peter Balfe[1], Aleem Siddiqui[2], Peter A.C. Wing[3], Jane A. McKeating[1,2]*

**1** Nuffield Department of Medicine, University of Oxford, United Kingdom, **2** Department of Medicine, University of California, California, United States of America, **3** Chinese Academy of Medical Sciences Oxford Institute, University of Oxford, United Kingdom

\* senko.tsukuda@ndm.ox.ac.uk (ST); jane.mckeating@ndm.ox.ac.uk (JAM)

**Data Availability Statement:** Primary data is available through Mendeley Data: Tsukuda, Senko (2024), "The N6-methyladenosine demethylase ALKBH5 regulates the hypoxic HBV

## Abstract

Chronic hepatitis B is a global health problem and current treatments only suppress hepatitis B virus (HBV) infection, highlighting the need for new curative treatments. Oxygen levels influence HBV replication and we previously reported that hypoxia inducible factors (HIFs) activate the basal core promoter (BCP). Here we show that the hypoxic-dependent increase in BCP-derived transcripts is dependent on $N^6$-methyladenosine ($m^6$A) modifications in the 5' stem loop that regulate RNA half-life. Application of a probe-enriched long-read sequencing method to accurately map the HBV transcriptome showed an increased abundance of pre-genomic RNA under hypoxic conditions. Mapping the transcription start sites of BCP-RNAs identified a role for hypoxia to regulate pre-genomic RNA splicing that is dependent on $m^6$A modification. Bioinformatic analysis of published single cell RNA-seq of murine liver showed an increased expression of the RNA demethylase ALKBH5 in the peri-central low oxygen region. *In vitro* studies with a human hepatocyte derived HepG2-NTCP cell line showed increased ALKBH5 gene expression under hypoxic conditions and a concomitant reduction in $m^6$A-modified HBV BCP-RNA and host RNAs. Silencing the demethylase reduced the level of BCP-RNAs and host gene (*CA9, NDRG1, VEGFA, BNIP3, FUT11, GAP* and *P4HA1*) transcripts and this was mediated via reduced HIFα expression. In summary, our study highlights a previously unrecognized role for ALKBH5 in orchestrating viral and cellular transcriptional responses to low oxygen.

## Author summary

Oxygen levels influence HBV replication and hypoxia inducible factors (HIFs) activate HBV transcription. Long-read sequencing and mapping the HBV transcriptome showed an increased abundance of viral RNAs under hypoxic conditions that was dependent on $N^6$-methyladenosine modifications. Investigating the oxygen-dependent expression of RNA demethylases identified ALKBH5 as a hypoxic activated gene and silencing its

transcriptome.", Mendeley Data, V1, doi: 10.17632/ydwvgx9yfw.1

**Funding:** The McKeating laboratory is funded by a Wellcome Investigator Award 200838/Z/16/Z, Wellcome Discovery Award 225198/Z/22/Z, Chinese Academy of Medical Sciences Innovation Fund for Medical Science, China (grant number: 2018-I2M-2-002). AM is supported by the John Black Foundation and AS is supported by NIH AI139234 grant. The funders had no role in study design, data collection and analysis, decision to publish, or preparation of the manuscript

**Competing interests:** The authors have declared that no competing interests exist.

expression showed a key role in regulating HBV and host gene expression under hypoxic conditions.

## Introduction

Chronic hepatitis B (CHB) is one of the world's most economically important diseases, with 2 billion people exposed to the virus during their lifetime resulting in a global burden of >290 million infections. Hepatitis B virus (HBV) replicates in the liver and chronic infection can result in progressive liver disease, cirrhosis and hepatocellular carcinoma (HCC) [1]. HBV is the prototypic member of the *hepadnaviridae* family of small enveloped hepatotropic viruses with a partial double-stranded relaxed circular DNA (rcDNA) genome. Current treatments include nucleos(t)ide analogs and interferons that suppress virus replication but are not curative largely due to the persistence of episomal HBV genomes and dysfunctional viral-specific immune responses [2].

HBV infects hepatocytes and the rcDNA genome translocates to the nucleus and is converted to covalently closed circular DNA (cccDNA) by host-DNA repair enzymes [3]. Several members of the host DNA repair pathway convert rcDNA to cccDNA that serves as the transcriptional template for viral RNAs [4]. HBV transcribes six major RNAs of decreasing length with a common 3' polyadenylation signal that include: pre-core (pC) that encodes e antigen (HBeAg); pre-genomic (pgRNA) that is translated to yield core protein and polymerase; preS1, preS2 and S RNAs encoding the surface envelope glycoproteins and the X transcript for the multi-functional x protein (HBx). Transcription is directed by four promoters that are regulated by two discrete but closely associated enhancer elements: Enhancer I activates the basal core promoter (BCP) which transcribes pC and pgRNAs, while Enhancer II activates promoters regulating the envelope and X RNAs [5,6]. pgRNA is encapsidated and reverse-transcribed by the viral polymerase to generate rcDNA genomes that can be enveloped and secreted as infectious particles [7]. Additional spliced isoforms of pC/pgRNA and preS2/S RNAs have been reported in experimental HBV replication models [8,9] and in liver biopsy samples from subjects with CHB [10], where the spliced variant 1 (SP1) is the most abundant and has been reported to associate with more progressive liver disease [11–13]. The spliced RNAs can encode fusion proteins that have the potential to influence viral replication [11,14,15] (reviewed in [16]). Our understanding of the interplay between HBV promoters and the host pathways that regulate RNA splicing are not well understood.

$N$6-methyladenosine (m$^6$A) is the most abundant modification found on eukaryotic transcripts where it can regulate mRNA structure, alternative splicing, stability and degradation [17]. m$^6$A modifications are regulated by the balanced activities of m$^6$A "writer" and "eraser" proteins. Adenosine is methylated by writers including the methyltransferase-like 3 (METTL3)/METTL14 complex [18] along with cofactors such as Wilms tumor 1-associated protein (WTAP) [19]. This complex typically methylates adenosine residues within the consensus DRACH (D = A, G or U; R = G or A; H = A, C or U) motif which is often located near stop codons, 3' untranslated regions and internal exons in mRNAs [20, 21]. m$^6$A modifications can be removed by erasers including the demethylases, AlkB Homolog 5 (ALKBH5) and fat mass and obesity-associated protein (FTO) [22,23]. HBV cccDNA encodes a DRACH motif present in all viral transcripts near the common 3' polyadenylation signal in a region termed the epsilon stem loop, but notably is also found in the 5' terminal repeat at the start of the pC and pgRNA and associated spliced transcripts [24]. m$^6$A modified HBV RNAs are recognized by YTH domain containing protein 2 (YTHDF2) and the interferon-induced RNase, ISG20,

that can process them for degradation [25]. More recently, m6A modified pgRNA was reported to be preferentially encapsidated [26,27]. Collectively, these studies demonstrate an important role for these post-transcriptional modifications at multiple stages of the HBV life cycle.

Oxygen concentration varies across different tissues, with the liver receiving oxygenated blood from the hepatic artery and partially oxygen-depleted blood via the hepatic portal vein, resulting in an oxygen gradient of 8–3% across the periportal and pericentral areas, respectively [28]. This oxygen gradient associates with liver zonation, a phenomenon where hepatocytes show distinct functional and structural organization across the liver [29]. Cells adapt to low oxygen through an orchestrated transcriptional response regulated by hypoxia inducible factors (HIFs): a heterodimeric transcription factor complex comprising one alpha (HIF-1α, HIF-2α and the less well studied HIF-3α) and one beta isoform (HIF-1β/ARNT). When oxygen is abundant, HIFα subunits are hydroxylated by HIF prolyl-hydroxylase domain (PHD) enzymes and targeted for proteasomal degradation. Under low oxygen conditions HIFs regulate a wide range of transcriptional targets that impact cellular processes (reviewed in [30]). HIFs can also be induced by non-hypoxic stimuli, such as immune, metabolic, and inflammatory signals [31], allowing flexible responses to diverse physiological stimuli [31–33]. We reported that HIFs bind and activate HBV cccDNA transcription both in laboratory models maintained under low oxygen and in HBV transgenic mice [34]. HIFs also suppress HBV cccDNA deamination by Apolipoprotein B mRNA Editing Catalytic Polypeptide-like 3B (APOBEC3B) [35] and potentiate virus replication. HIFs have been reported to influence the replication of a number of viruses (reviewed in [36]), enhancing Epstein Barr virus replication [37,38] but suppressing Influenza A and SARS-CoV-2 [39–41], highlighting the complex interplay between hypoxia signalling and viral infection.

To date, studies investigating the role of m6A modifications in the HBV life cycle have been performed under standard laboratory conditions of 18% oxygen, where HIFs are inactive. Several reports show that HIFs activate the m6A demethylase ALKBH5 [42–44] and we hypothesize that low oxygen conditions of the liver will increase ALKBH5 expression and influence the methylation status of HBV RNAs and their abundance in the infected cell. Our study identifies a role for m6A modifications in the 5' epsilon stem loop in the regulation of pC and pgRNA half-life and splicing under hypoxic conditions. Furthermore, we identify a role for ALKBH5 in the regulation of HIF-1α under hypoxic conditions that impacts the abundance of HBV and cellular transcripts.

## Results

### Hypoxic increase in HBV basal core promoter derived transcripts is dependent on m6A modifications

To assess the role of m6A RNA modifications in regulating the steady-state levels of HBV RNA under low oxygen conditions we mutated the DRACH motifs to generate m6A-null virus as previously reported (**Fig 1A**) [24]. Culturing uninfected, HBV wild type (WT) or m6A-null infected cells under 1% oxygen stabilised HIF-1α and HIF-2α expression. To measure HIF-specific regulation of HBV transcripts we treated the infected cells with a HIF prolyl hydroxylase inhibitor (FG-4592) that can stabilize HIFs under normoxic conditions [45] (**Fig 1A**). Both treatments induced gene expression of the HIF-target gene carbonic anhydrase 9 (*CA9*) [46] in WT and m6A-null infected samples (**Fig 1A**). To quantify HBV transcripts we selected primers targeting the 5' end of the genome to amplify BCP-derived transcripts (pC, pg and their spliced derivatives, SP) or the 3' region shared by all transcripts (total HBV RNA) (**Fig 1B**). Low oxygen or FG-4592 treatment increased the level of total viral RNAs in WT or m6A-null infected cells, however BCP-derived transcript levels in the m6A-null infected cells were

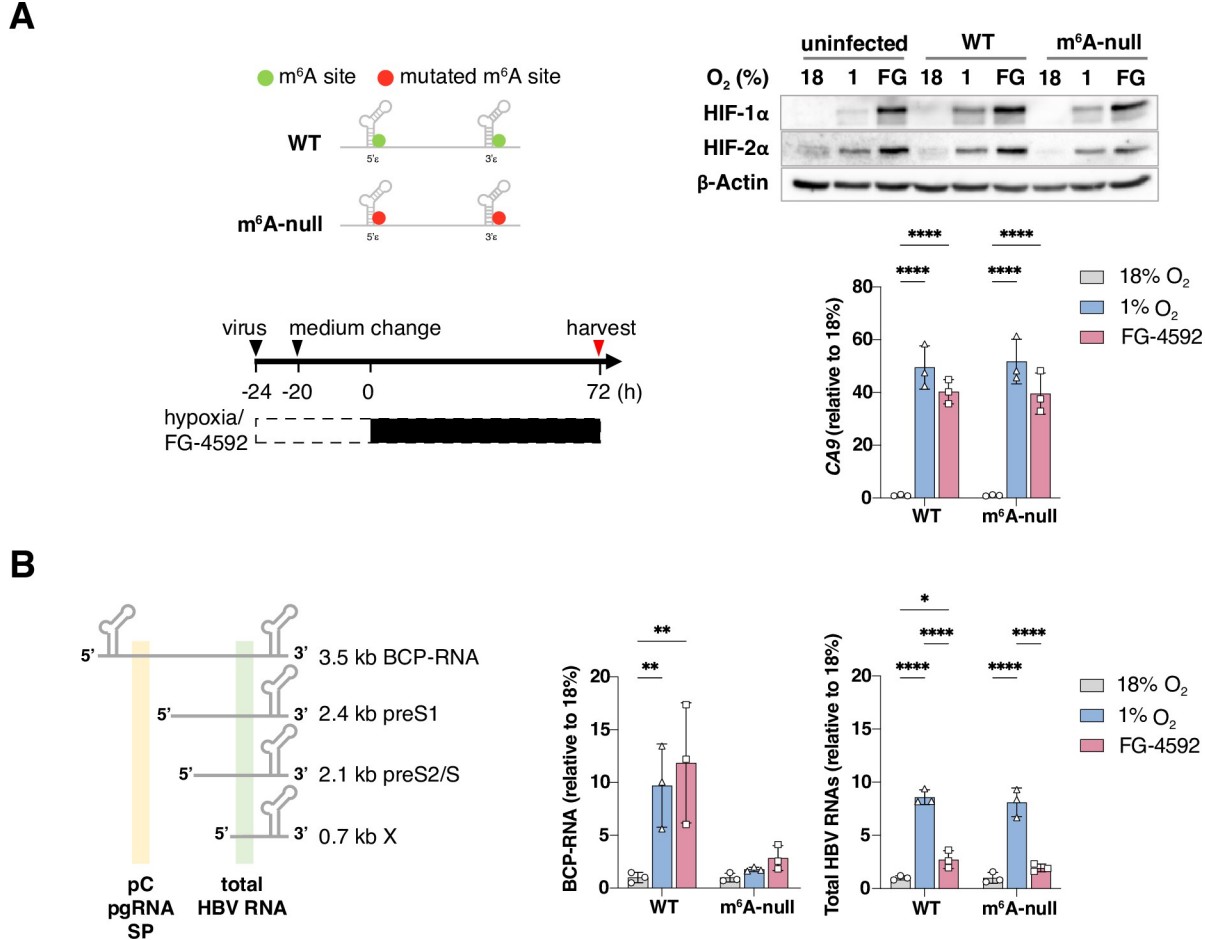

**Fig 1. Hypoxic regulation of HBV BCP-RNAs is dependent on m⁶A modification.** (**A**) Schematic of DRACH mutations in 5' and 3' HBV RNA stem loops and protocol for virus infection. HepG2-NTCP cells were infected with HBV wild type (WT) or m⁶A-null and cultured at 18%, 1% oxygen or treated with FG-4592 (FG, 30 μM) for 72h. HIF-1α, HIF-2α, or β-Actin protein levels and HIF-regulated gene carbonic anhydrase 9 (*CA9*) transcripts were measured. (**B**) Schematic of primer regions used for qRT-PCR amplification of BCP-RNAs (pC, pgRNA and spliced RNAs, SP) and total HBV RNAs. Data are expressed relative to 18% oxygen for each condition and presented as mean ± S.D. of n = 3 from one of three independent experiments with statistical significance determined using a two-way ANOVA. * $p < 0.05$, ** $p < 0.01$, **** $p < 0.0001$.

unchanged (**Fig 1B**). Together, these data suggest an essential role for m⁶A modifications in the hypoxic induction of BCP-RNAs.

## The 5' stem-loop DRACH motif regulates the abundance of BCP-RNAs under hypoxic conditions

To investigate which m⁶A motif regulates BCP-RNA levels under low oxygen conditions, we transfected HepG2-NTCP cells with HBV 1.3x overlength plasmids encoding mutations in the DRACH motif either in the 5' stem-loop (HBV m⁶A-5'null), the 3' stem-loop (HBV m⁶A-3'null) or both loops (HBV m⁶A-null) (**Fig 2A**). Transfected HepG2-NTCP cells were cultured at 18% or 1% oxygen conditions and BCP-RNA, HBeAg antigen, core protein and secreted HBV DNA measured. Measuring intracellular HBV DNA at 4h post transfection demonstrated similar efficiencies of plasmid uptake (**S1A Fig**) and we confirmed the cellular hypoxic response by measuring HIF protein levels and *CA9* gene expression (**Figs 2A** and **S1B**). Lower

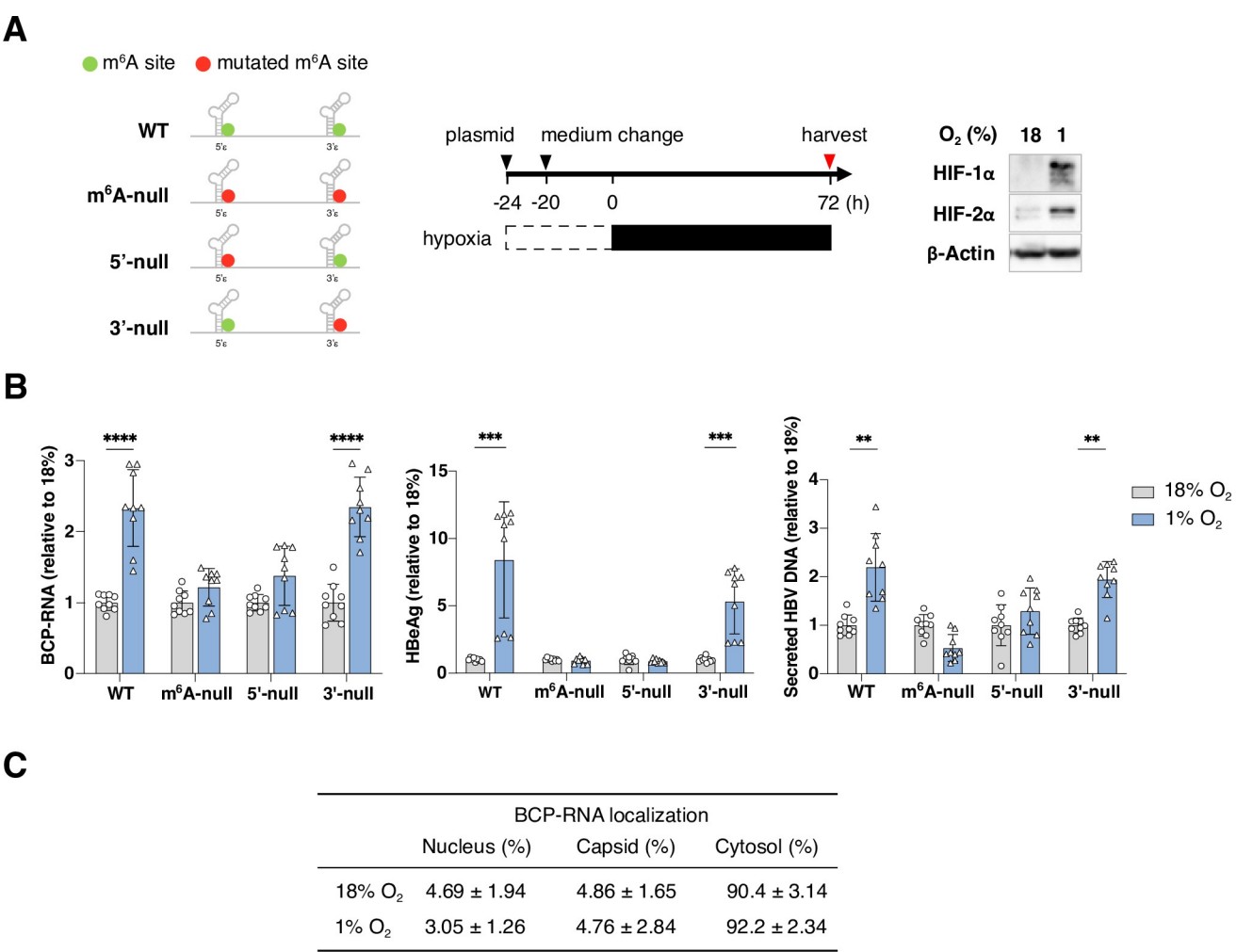

**Fig 2. Hypoxic dependent increase in HBV BCP-RNAs and secreted viral DNA is dependent on 5' m⁶A modification.** (**A**) Schematic of HBV plasmids indicating the location of the A-C mutation of the 5' and 3' m⁶A sites in BCP-RNAs. Circles represent WT (green) and the C mutation (red). HepG2-NTCP cells were transfected with HBV plasmids for 4h and incubated for an additional 20h at 18% oxygen, before transferring to 18% or 1% oxygen for 72h. HIF-1α, HIF-2α, and β-Actin protein expression was assessed. (**B**) Intracellular BCP-RNAs, extracellular secreted HBeAg and HBV DNA were quantified and data presented as mean ± S.D. of three independent experiments. Statistical significance was determined using Mann-Whitney tests, with Bonferroni correction for multiple comparisons. ** $p < 0.01$, *** $p < 0.001$, **** $p < 0.0001$. (**C**) HepG2-NTCP cells were prepared as shown in A and RNA extracted from the cellular fractions, treated with TURBO DNase to remove any contaminating plasmid DNA and BCP-RNA quantified by qRT-PCR. Data were obtained from n = 5 independent samples and presented as mean ± S.D.

levels of BCP-RNAs were noted in the m⁶A-null and m⁶A-5'null transfected cells compared to WT or m⁶A-3'null, consistent with our earlier report showing reduced replicative fitness (**S1C Fig**) [24]. Hypoxia increased the levels of BCP-RNAs, core protein and extracellular HBV DNA in the WT or m⁶A-3'null transfected cells, whilst neither m⁶A-null nor m⁶A-5'null showed any oxygen-dependent modulation. (**Figs 2B** and **S1D**)

To understand whether the hypoxia-associated increase in BCP-RNAs leads to gross changes in their cellular localization we transfected HepG2 cells with HBV and quantified the relative abundance of viral transcripts in the nucleus, cytosol or in viral capsids. The majority of BCP-RNAs were in the cytosol and this did not change under low oxygen conditions (**Fig 2C**), suggesting that hypoxia does not influence the encapsidation machinery and/or assembly

processes. In summary, these data show that 5' m$^6$A modification plays an essential role in the hypoxia-dependent increase in BCP-RNAs.

## Hypoxic increase in HBV BCP-RNA half-life is m$^6$A-dependent

As methylation and hypoxia can both influence RNA stability [47,48] we measured the half-life of WT or m$^6$A-null derived BCP-RNAs under low-oxygen conditions using a recently developed 'Roadblock PCR' [49] (**Fig 3A**). This assay enabled us to measure mRNA decay kinetics in HBV transfected cells without using the general transcription blocker actinomycin D that can affect cell viability. We observed a significant increase in the half-life of WT BCP-RNAs under hypoxic conditions (5.5h v 10.6h), that was not apparent in the m$^6$A-null transfected samples (8.6h v 9.1h) (**Fig 3B**). The half-life ($t_{1/2}$) was estimated from a one phase exponential decay with 95% confidence intervals (**Fig 3C**). In summary, these data highlight an essential role for m$^6$A modifications in regulating the half-life of BCP-RNAs under hypoxic conditions.

## Mapping the hypoxic HBV transcriptome and impact of m$^6$A modification

To further explore the interplay between hypoxia and methylation status of the HBV transcriptome we used our previously reported probe-enrichment long-read sequencing approach [9] to map unspliced and spliced transcripts (**Fig 4A**). We quantified viral reads originating at previously reported transcription start sites (TSSs) [8,50] in WT and m$^6$A-null transfected HepG2 cells. The sequenced libraries contained 47,697–119,071 reads and we noted a similar frequency of HBV reads among the samples, ranging from 41–55% of the total (**S1 Table**). To compare the profile of HBV RNAs in the different samples we expressed the viral reads as transcripts per million (TPM). pC and pgRNAs are a minor component of the viral transcriptome (<5%) with RNAs encoding the surface glycoproteins (preS1, preS2, S) being the most abundant transcripts irrespective of hypoxic conditions or methylation status (**Fig 4B**). A number of viral transcripts did not map to known TSS and were classified as incomplete, with many of these encoding an S gene open reading frame (ORF) (**S1 Table**). Mapping the spliced transcripts confirmed SP1 as the most abundant (~1,500–8,000 TPM) with SP9, SP6, SP7, SP14 and pSP9 transcripts also detected (>500 TPM) (**S1 Table**). Differential expression analysis identified a significant 3.9-fold increase in pgRNA whereas pC (pC-L and pC-S) transcripts showed a non-significant (1.3-fold) change, implying that our earlier observations of the increase in BCP-RNAs under hypoxia most likely reflect changes in pgRNA abundance. Modest increases in preS2 (1.8-fold) and S (1.9-fold) transcripts were observed in WT transfected cells cultured under hypoxic conditions (adjusted p-values $< 10^{-5}$), suggesting hypoxic potentiation of viral transcripts extending beyond the BCP (**Fig 4C**).

Several studies report an association between m$^6$A levels and alternative mRNA splicing [51,52] including the adenovirus transcriptome [53], although the underlying mechanisms were not elucidated. Long read sequencing identified the BCP-RNAs and we mapped two pC transcripts (pC-Long (L) or pC-Short (S)), pgRNA and a number of spliced RNAs which encode the 5' stem-loop DRACH motif (**Fig 4D** and **S1 Table**). As the majority of pC transcripts initiated at 210–270 nucleotides with pC-L transcripts being relative rare (6.1%), all subsequent analyses pooled these transcripts (**S1 Table**). Under normoxic conditions the spliced transcripts originated from pC and pgRNAs in WT and m$^6$A-null transfections (**Fig 4E**). However, there was a significant increase in the frequency of pgRNA-derived spliced transcripts in WT transfected samples under hypoxic conditions that was not apparent in the m$^6$A-null samples (**Figs 4F** and **S2**). These long-read sequencing studies highlight a role for

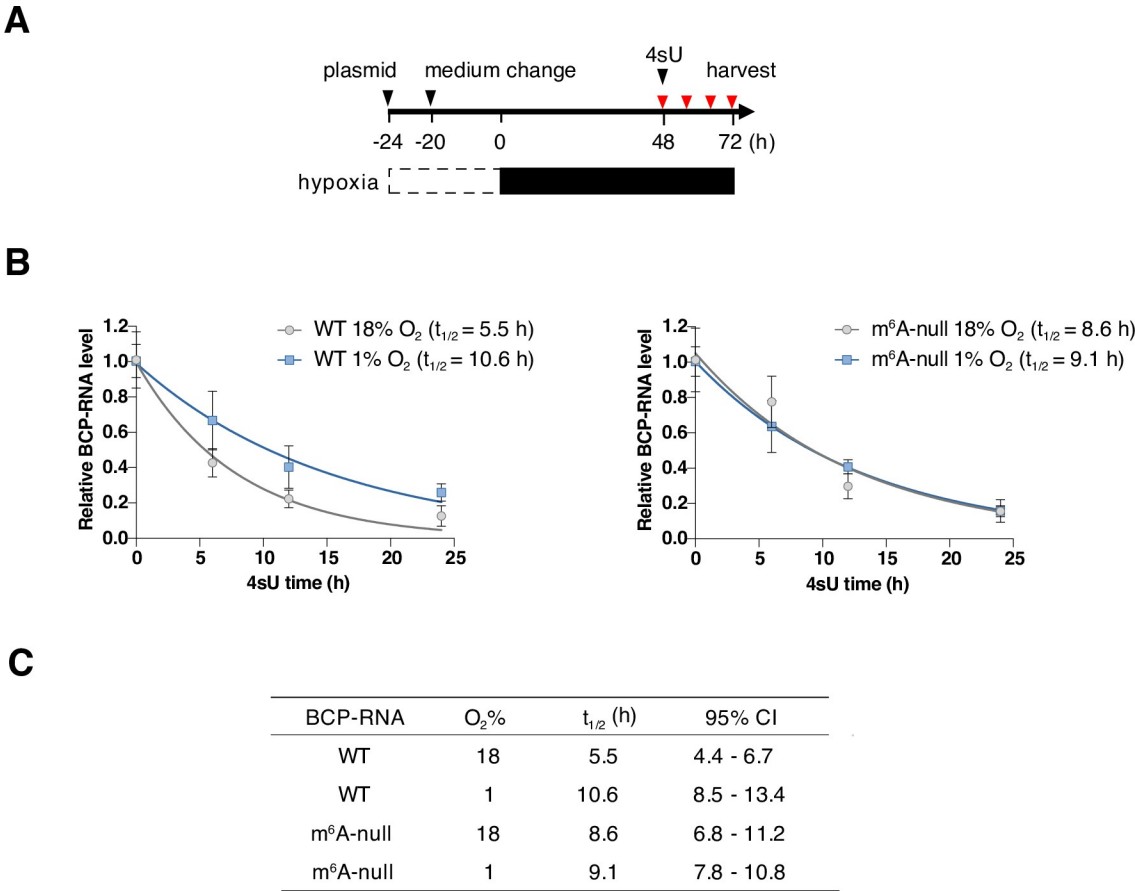

**Fig 3. Hypoxic increase in HBV BCP-RNA half-life is dependent on m⁶A-modification.** (**A**) Schematic of experimental procedure. HepG2-NTCP cells transfected with HBV WT or m⁶A-null plasmids were incubated at 18% or 1% oxygen conditions for 48h. Cultures were treated with 4-thiouridine (4sU) and cells harvested at 0, 6, 12, and 24h post 4sU treatment. (**B**) BCP-RNA levels were quantified by qRT-PCR and data presented relative to 0h 4sU treatment and represent the mean ± S.D. of n = 6 from two independent experiments. (**C**) Half-life ($t_{1/2}$) was analyzed by nonlinear Regression one phase exponential decay and the 95% confidence intervals (CI) listed from two independent infections.

m⁶A post-transcriptional modifications in regulating the hypoxic HBV transcriptome and the fate of pgRNA splicing.

## Hypoxic activation of ALKBH5 impacts m⁶A modified HBV and host transcripts

The m⁶A demethylase ALKBH5 is a direct target of HIF-1α and HIF-2α [42–44] and is hypoxic regulated in a variety of cell lines, including breast cancer and adipocytes. We investigated ALKBH5 expression in the liver using a published single cell RNA-seq data set from murine liver [54]. We observed an enrichment of *Alkbh5* transcripts in the low oxygen pericentral area (PC) compared to the perivenous region (PV) (**Fig 5A**). Zonal expression of the HIF target gene, N-myc downstream regulated 1 (*Ndrg1*) was noted, whereas expression of m⁶A demethylase Fat mass and obesity associated (*Fto*) gene was comparable across the liver lobule (**Fig 5A**). To ascertain whether these observations translate to our *in vitro* model we cultured HepG2 cells under 18% and 1% oxygen and showed an increase in ALKBH5 protein and gene expression in hypoxic cells, whereas *FTO* gene expression was unchanged (**Fig 5B–5C**). There was no evidence for hypoxic regulation of m⁶A writers (METTL3, METTL14, and WTAP) or

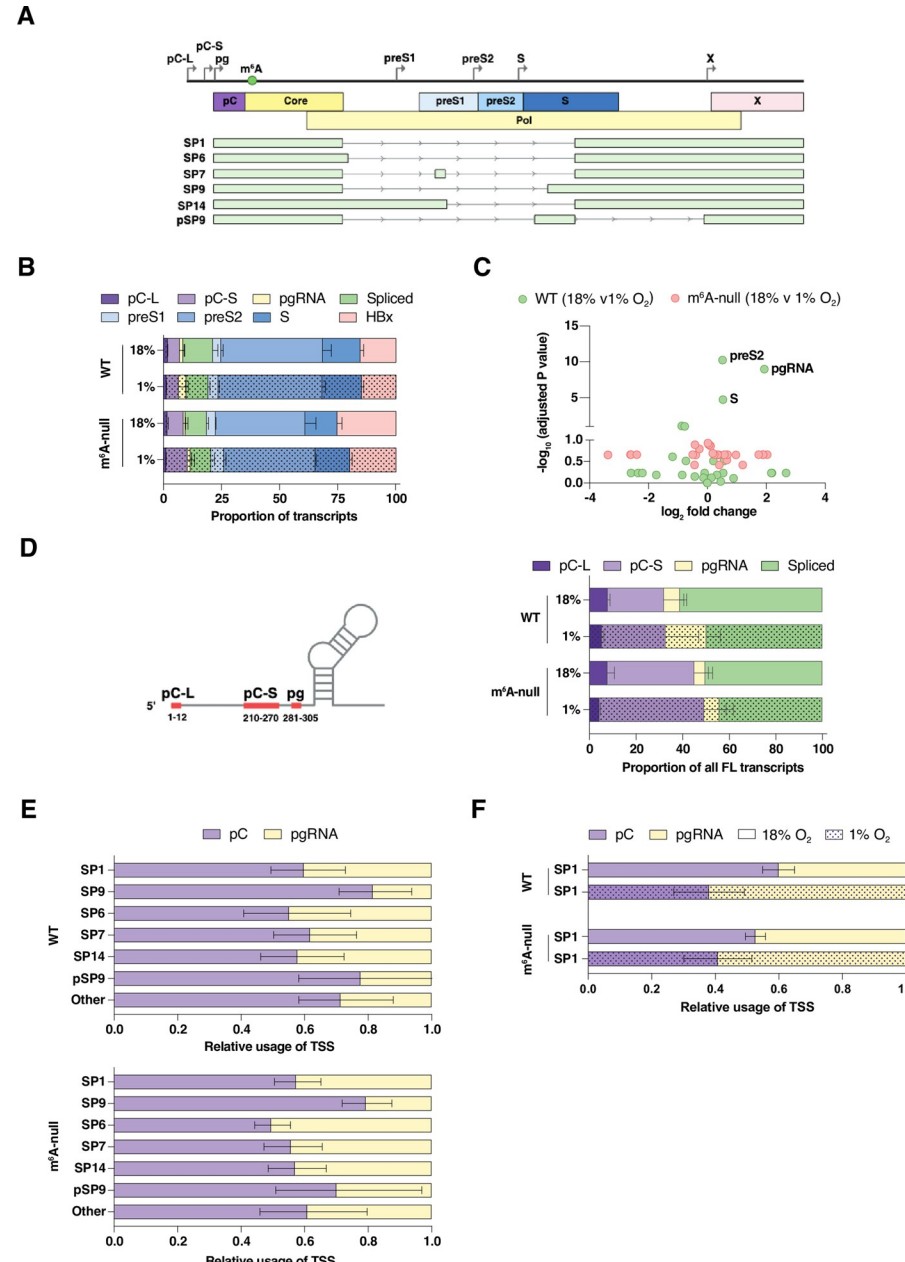

**Fig 4. HBV long-read sequence analysis.** (**A**) Cartoon depicting the major unspliced and spliced transcripts generated from the HBV genome, where the transcription start sites (TSS) for the major viral RNAs are shown. (**B**) Relative frequencies of major unspliced and spliced transcripts in HBV WT or m6A-null expressing HepG2 cells cultured under 18% or 1% oxygen. (**C**) Differential gene expression of hypoxic regulated HBV WT and m6A-null RNAs, where the X axis denotes differences between samples (Log2-Fold Change) and the Y axis the adjusted probability of the changes (-log10 (adjusted p-value)). The threshold for significance was defined as 5% (-log10[0.05] = 1.303). (**D**) Schematic depicting the range of TSS for BCP-RNAs (pC-L, pC-S, pg and associated spliced RNAs) and their relative frequency in HBV WT or m6A-null transfected cultures at 18% or 1% oxygen. (**E**) Relative TSS usage of BCP-derived spliced RNAs in HBV WT or m6A-null transfections (18% O2) with transcripts listed in order of abundance (see S1 Table). (**F**) Relative TSS usage of SP1 RNA in HBV WT or m6A-null transfected cultures at 18% or 1% oxygen. In all panels the error bars denote standard deviations from the mean TPM of the replicates listed in S1 Table.

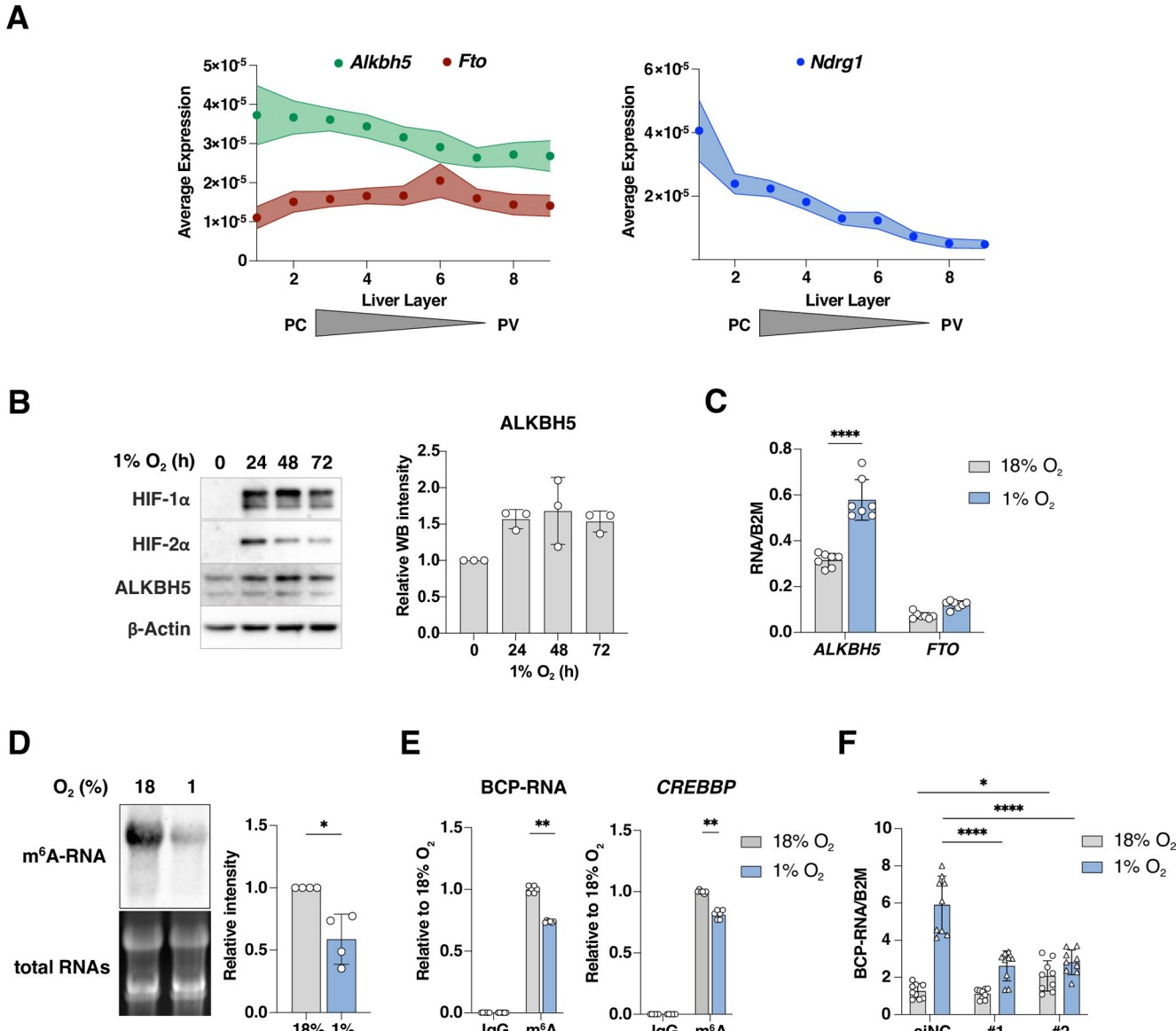

**Fig 5. Hypoxic increase in ALKBH5 expression regulates the methylation and abundance of HBV BCP-RNAs.** (**A**) Zonation of the RNA demethylases ALKBH5, FTO and NDRG1 based on scRNA-seq data from mouse liver [54]. (**B**) HepG2-NTCP cells were cultured under 18% or 1% oxygen for the indicated times and HIF-1α, HIF-2α, ALKBH5 or β-Actin protein expression measured. ALKBH5 expression was quantified by densitometry. (**C**) *ALKBH5* and *FTO* transcript levels in HepG2-NTCP cells cultured under 18% or 1% oxygen for 72h. Data are expressed relative to a housekeeping gene *B2M* and presented as mean ± S.D. of n = 4 from 2 experiments. Statistical significance was determined using Mann-Whitney tests, with Bonferroni correction for multiple comparisons, **** $p < 0.0001$. (**D**) m6A-modified cellular RNAs. RNA was extracted from HepG2-NTCP cells cultured under 18% or 1% oxygen conditions for 72h and subjected to immuno-northern blotting using m6A antibody. Densitometric quantification of northern blots was performed and data expressed relative to 18% oxygen. Data is shown as the mean ± S.D. of 4 independent experiments with statistical significance determined using Mann-Whitney test. * $p < 0.05$. (**E**) Quantification of methylated HBV BCP-RNAs. HepG2-NTCP cells were transfected with HBV WT plasmid for 18h and cultured at 18% or 1% oxygen for 72h. Total cellular RNA was extracted and subjected to a Methylated RNA immunoprecipitation (MeRIP) assay and BCP-RNA and control *CREBBP* RNA measured by qPCR. Data are presented as the mean ± S.D. of n = 6 samples from 3 independent experiments with statistical significance determined using Mann-Whitney tests with Bonferroni correction for multiple comparisons. ** $p < 0.01$. (**F**) HepG2-NTCP cells were transfected with siRNAs (NC; Non-targeting, ALK; ALKBH5) for 6h, followed by delivery of HBV WT plasmid and cultured at 18% or 1% oxygen for 72h. RNA was extracted and BCP-RNA levels quantified by qRT-PCR. Data are expressed relative to a house keeping gene *B2M* and presented as the mean ± S.D. of n = 9 samples from 3 independent experiments. Statistical significance was determined using Mann-Whitney tests with Bonferroni correction for multiple comparisons. * $p < 0.05$, **** $p < 0.0001$.

m6A reader proteins (YTHDF1-3) (**S3 Fig**). To assess the functional consequences of the increased ALKBH5 expression, we measured total m6A-RNA levels by immuno-northern blotting with an anti-m6A antibody and observed a reduction in m6A-modified cellular RNAs under hypoxic conditions, consistent with the increase in demethylase expression (**Fig 5D**).

To investigate whether hypoxia alters BCP-RNA methylation status we used an m6A–RNA immunoprecipitation (MeRIP) assay combined with qPCR and showed a significant reduction in precipitated viral RNA under hypoxic conditions (**Fig 5E**). As a control we PCR quantified the m6A modified CREBBP transcript and noted a reduction under hypoxic conditions (**Fig 5E**). To investigate the potential role of ALKBH5 in regulating the abundance of BCP-RNAs we silenced the demethylase with two independent siRNAs targeting different exons. We confirmed siRNA efficacy in cells cultured at 18% or 1% oxygen with siRNA #2 that showed a greater inhibition of ALKBH5 expression (**S4 Fig**). Silencing ALKBH5 with either siRNA blunted the hypoxic-associated increase in BCP-RNAs (**Fig 5F**), suggesting a role for this demethylase in regulating these viral RNAs under low oxygen conditions.

### ALKBH5 regulates HIF-α expression

As we had previously identified a role for HIFs to bind HBV cccDNA and activate transcription, we investigated the interplay between ALKBH5 and HIFs. *HIF-1α* mRNA is methylated in HepG2 cells and comparable levels of anti-m6A precipitated transcripts were noted under hypoxic conditions (**S5A Fig**). HepG2 cells were transfected with siRNAs targeting ALKBH5 or an irrelevant negative control (NC) and cultured at 1% oxygen for 24h to stabilize HIF expression and activate NDRG1 expression. Cells transfected with siALKBH5 showed reduced levels of both HIF-α isoforms and NDRG1, whereas comparable levels of HIF-1α and HIF-2α transcripts were noted in the silenced cells (**Fig 6A–6B**). Silencing ALKBH5 in HepG2 cells blunted the hypoxic induction of a panel of HIF target genes (*CA9*, *NDRG1*, *VEGFA*, *BNIP3*, *FUT11*, *GP1*, and *P4HA1*), demonstrating the broad impact of this demethylase to regulate HIF-transcriptional activity (**Fig 6C**).

When oxygen is abundant HIF-α is hydroxylated by PHD enzymes, leading to its ubiquitination and proteasomal degradation. ALKBH5 silencing had no major effects on the levels of hydroxylated HIF-1α (**S6A Fig**). Recent reports show that histone deacetylase 4 (HDAC4) regulates HIF-1α protein acetylation and stability [55–57] and ALKBH5 regulates m6A-modification of *HDAC4* transcripts under hypoxic conditions in pancreatic cancer cells [58]. To investigate whether ALKBH5 regulates *HDAC4* m6A modifications in HepG2-NTCP hepatoma cells we precipitated methylated *HDAC4* mRNA in ALKBH5 silenced cells under hypoxic conditions. We noted a significant increase in the level of anti-m6A precipitated *HDAC4* transcripts in the silenced hypoxic cells (**S6B Fig**). To assess the role for ALKBH5 in post-transcriptional regulation of HIF-α, HepG2 cells were transfected with siALKBH5 #2 as it showed the most robust silencing. The cells were cultured under hypoxic conditions for 24h before treatment with cycloheximide (CHX) to inhibit protein synthesis. CHX treatment inhibited HIF-1α and HIF-2α expression and their degradation was accelerated in the absence of ALKBH5 (**Fig 6D**). Furthermore, both HIF-α isoforms was reduced in ALKBH5-depleted cells treated with the proteasomal inhibitor MG132 (**Fig 6D**). Collectively, these data illustrate a role for the ALKBH5 demethylase in regulating HIFα expression and function.

### Discussion

Our study shows an essential role for m6A modifications in regulating the half-life of BCP-RNAs under hypoxic conditions. Long-read sequencing of the infected cells allowed us to accurately map the BCP-RNAs and we observed a significant increase in pgRNA under

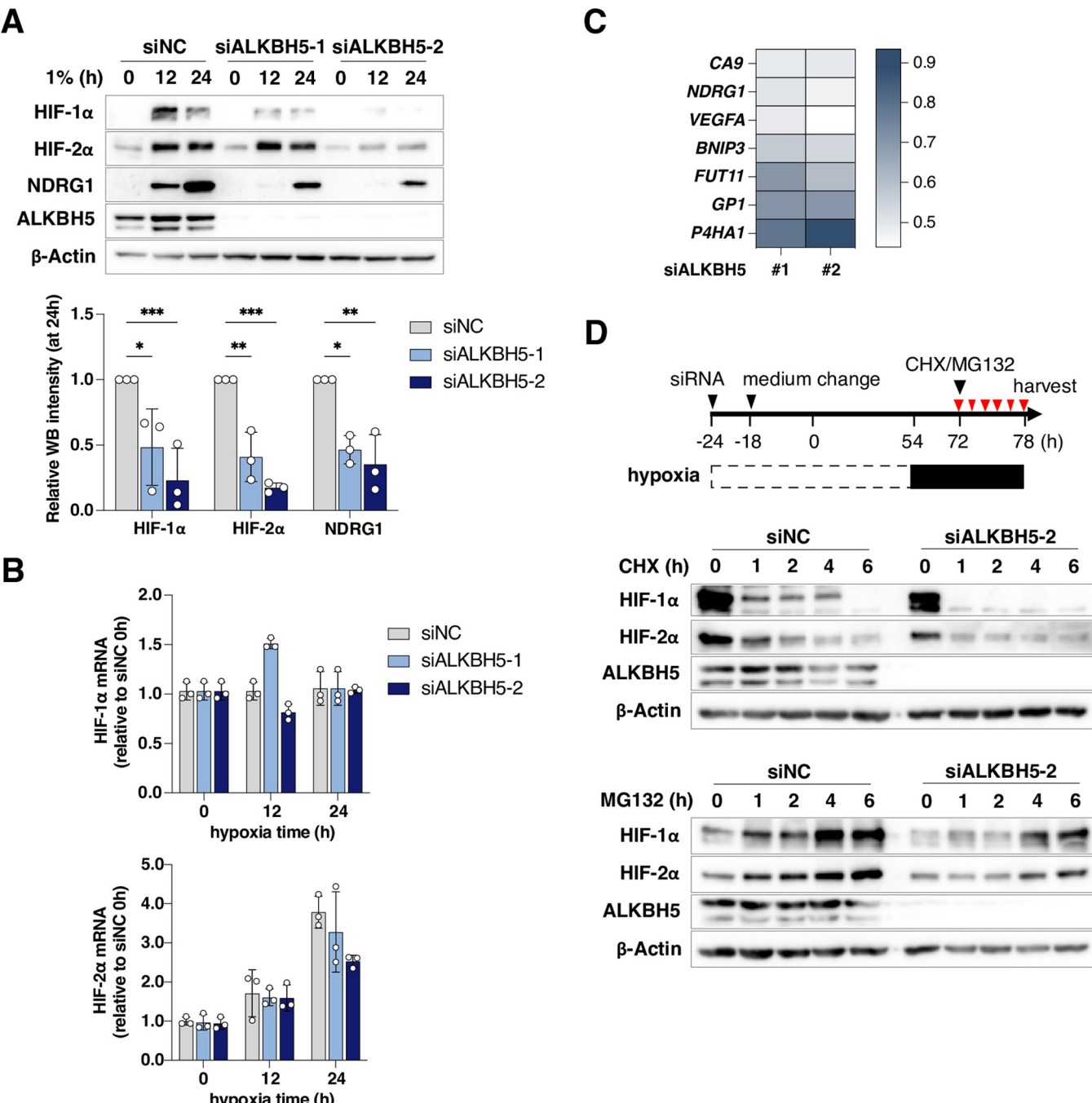

**Fig 6. ALKBH5 regulates HIFα expression under hypoxic conditions.** (**A**) HIF-α expression in ALKBH5 silenced cells. HepG2-NTCP cells were transfected with ALKBH5 specific siRNAs for 48h and cultured under 1% oxygen for 12 or 24h. Samples were probed for HIF-1α, HIF-2α, NDRG1, ALKBH5 or β-Actin proteins and expression quantified by densitometry. Data is expressed relative to 24h hypoxic siNC samples and represents the mean ± S.D. from 3 independent experiments. Statistical significance was determined using a two-way ANOVA. * $p < 0.05$, ** $p < 0.01$. (**B**) *HIF-1α* and *HIF-2α* mRNAs were measured by qRT-PCR and data presented relative to siNC hypoxic conditions. (**C**) Expression of HIF-regulated genes in ALKBH5 silenced cells. HepG2-NTCP cells were cultured as listed in Fig 5E and the indicated gene transcripts quantified by qRT-PCR with data expressed relative to siNC under hypoxic conditions. Data represent the mean ± S.D. of samples from 3 independent experiments. (**D**) HIF-1α protein expression in ALKBH5 silenced cells. HepG2-NTCP cells were cultured in 1% oxygen for 24h and treated with cycloheximide (CHX, 20 μg/mL) or MG132 (20 μM) for the indicated times. Samples were collected and probed for HIF-1α, HIF-2α, ALKBH5 and β-Actin by western blotting. Note: Different exposure times were used for the upper and lower panels.

hypoxic conditions in the WT infected cells that was not apparent in the m⁶A-null samples. These long-read sequencing studies highlight a role for $m^6A$ post-transcriptional modifications in regulating the fate of pgRNA splicing. Given the overlapping nature of the HBV transcripts we cannot discriminate pC and pgRNA by qPCR; the MeRIP assays confirmed that both transcripts are methylated. The observation that the hypoxia-mediated increase in pgRNA is dependent on 5'-$m^6A$ modification suggests specific recognition mechanisms. Wang *et al* analyzed the location of $m^6A$ modifications on RNAs isolated from hypoxic HeLa cells and reported that cells undergo a reprogramming of their $m^6A$ epitranscriptome by altering both the $m^6A$ modification at specific sites and their distribution patterns [48]. We cannot exclude that hypoxia may alter other $m^6A$ modified sites on HBV RNAs. Mutation of the 5'-$m^6A$ DRACH HBV motif at position 1907 (A to C) is unlikely to impair HIF-1α binding to hypoxic responsive elements that are located in the BCP at residues 1751–1769. Further experiments to image the intracellular location of methylated and non-methylated transcripts in infected cells may provide mechanistic insights.

Recent studies have highlighted an interplay between hypoxia signalling and $m^6A$-post-transcriptional regulatory pathways; with decreased levels of $m^6A$-RNAs seen in breast cancer, cardiac microvascular endothelial and cervical cancer cells [43,59]. In contrast, increased levels of $m^6A$ modified RNAs were found in human umbilical vein endothelial cells, cardiomyocytes and murine hearts under hypoxic conditions that associated with increased methyltransferase METTL3 expression [60–62]. Collectively, these studies show that hypoxic modulation of $m^6A$-RNA is dependent on both the tissue and cell type. Consistent with our observations, Wang *et al* reported a hypoxic reduction in $m^6A$ modified RNAs in human hepatoma Huh-7, HepG2 and Hep3B cell lines that was partially reversed by silencing ALKBH5 [48]. Reports that $m^6A$ modified RNAs are perturbed in inflammatory diseases highlight a potential therapeutic role for targeting this pathway (reviewed in [63]).

Analyzing a published single-cell RNA sequencing (scRNA-seq) data set from mouse liver [54] showed a zonation of *Alkbh5* but not *Fto*, with transcripts more abundant in the pericentral region of the liver, consistent with hypoxic regulation. ALKBH5 is a HIF regulated gene and increased expression is noted under low oxygen conditions. ALKBH5 activity is also regulated by post-translation methylation and SUMOylation modifications that provide a further level of control over demethylase activity [64]. Our results showing reduced BCP-RNAs levels in HBV infected ALKBH5 silenced cells under hypoxic conditions supports a positive role for this demethylase in regulating susceptibility to viral infection. This conclusion is consistent with our earlier work that reported an increased expression of HBV antigen expressing cells in the pericentral 'high ALKBH5' areas of the liver in HBV transgenic mice [34,35]. Liu *et al* reported that *Alkbh5* deficient mice were resistant to infection by a range of DNA and RNA viruses (VSV, HSV-1 EMCV) mediated by an $m^6A$ RNA-dependent down regulation of the α-ketoglutarate dehydrogenase (OGDH)-itaconate pathway that promoted virus replication [65]. The authors reported that Vesicular Stomatitis Virus infection targeted ALKBH5 to evade this host-restriction pathway. We have limited evidence of HBV infection altering ALKBH5 expression or activity in our experiments. Qu *et al* reported that HBx increased H3K4me3 modifications in the ALKBH5 promoter region that associated with increased demethylase expression [66]. The authors noted increased ALKBH5 expression in HBV-HCC, however, there were no mechanistic studies to link this directly to HBx and, given the hypoxic nature of HCC (reviewed in [67]), could align with a HIF-driven activation of ALKBH5 gene expression.

Adenosine methylation can regulate many aspects of mRNA metabolism including splicing, nuclear export, stability, and translation. Most translation events in the cell occur through recognition of the cap by the eIF4F protein complex. However, cellular states such as apoptosis,

mitosis or the stress response can suppress cap-dependent translation allowing selected mRNAs to be translated. Recent studies have reported that stress conditions, such as heat shock or amino acid starvation, promote nuclear trafficking of the $m^6A$ reader proteins YTHDF1 and YTHDF2 [68,69] and increase cap-independent translation of mRNAs from an $m^6A$-modified 5'UTR [68,70–72]. We found that ALKBH5 silencing reduced HIF-α expression without affecting RNA levels. Silencing of ALKBH5 did not alter the levels of hydroxylated HIF-1α, suggesting that this demethylase does not directly regulate PHD activity, however, functional enzymic studies would be required to address this possibility. We show increased $m^6A$-modifications of *HDAC4* in ALKBH5 silenced cells that may influence HIF-α protein stability in our hepatoma model as previously reported in pancreatic cancer [58]. There is a report showing that PBMR1, a component of the chromatin remodeler SWI/SNF, positively regulates HIF-1α translation through protein interaction with YTHDF2 under normoxic and hypoxic conditions and HIF expression was reduced when these proteins were silenced [68]. These reports are consistent with an essential role for $m^6A$ modified RNAs in cellular adaption to hypoxic stress. Taken together, our results support a role for ALKBH5 in regulating HIF-1α stability and transcriptional activity and in cellular adaption to hypoxic stress. Our findings on the cross-talk of ALKBH5 and HIF signalling provide mechanistic insights into cellular responses that regulate HBV transcripts and may be more widely applicable to other liver tropic pathogens (**Fig 7**).

## Materials and methods

### Reagents

FG-4592 was obtained from either Selleckchem or MedChemExpress. Cyclohexamide was purchased from Abcam. siRNAs were obtained from Thermo Fisher Scientific, MG132 and 4-thiouridine, and N-ethylmaleimide were purchased from Sigma-Aldrich. Primary antibodies used in this study are ALKBH5 (Atlas Antibodies), β Actin (Sigma), HIF-1α (BD Bioscience), HIF-2α (NOVUS), HIF-1β (NOVUS), NDRG1 (CST), HBc (DAKO), anti-$m^6A$ polyclonal antibody (Synaptic Systems). HRP-conjugated secondary antibodies were purchased from DAKO.

### Cell lines

HepG2-NTCP cells were maintained in Dulbecco's Modified Eagles Medium (DMEM) (ThermoFisher), containing Glutamax supplemented with 10% Fetal Bovine Serum, 50 U/ml Penicillin/Streptomycin, and non-essential amino acids. Cells were maintained in 5% $CO_2$ and 18% oxygen. Hypoxic treatment of cells was carried out in a hypoxic incubator (New Brunswick Galaxy 48R, Eppendorf) or a hypoxia chamber (InvivO₂, Baker-Ruskinn Technologies) at 5% $CO_2$ and 1% oxygen.

### HBV preparation and infection

HBV was prepared from the supernatant of HepG2 cells transfected HBV1.3 plasmids [24] by polyethylene glycol 8000 precipitation. Briefly, culture media was mixed with 10% PEG8000/ 2.3% NaCl and incubated at 4°C for 16h, then centrifuged at 4500 rpm for 1h at 4°C. After discarding the supernatant, the pellet was resuspended in DMEM (~200-fold concentration). The inoculum was treated with DNase at 37°C for 1h, DNA extracted and quantified by qPCR to measure HBV genome copies. HepG2-NTCP cells were seeded on collagen coated plasticware and infected with HBV (MOI of 300 or 1,000 based on genome copies) in the presence of 4%

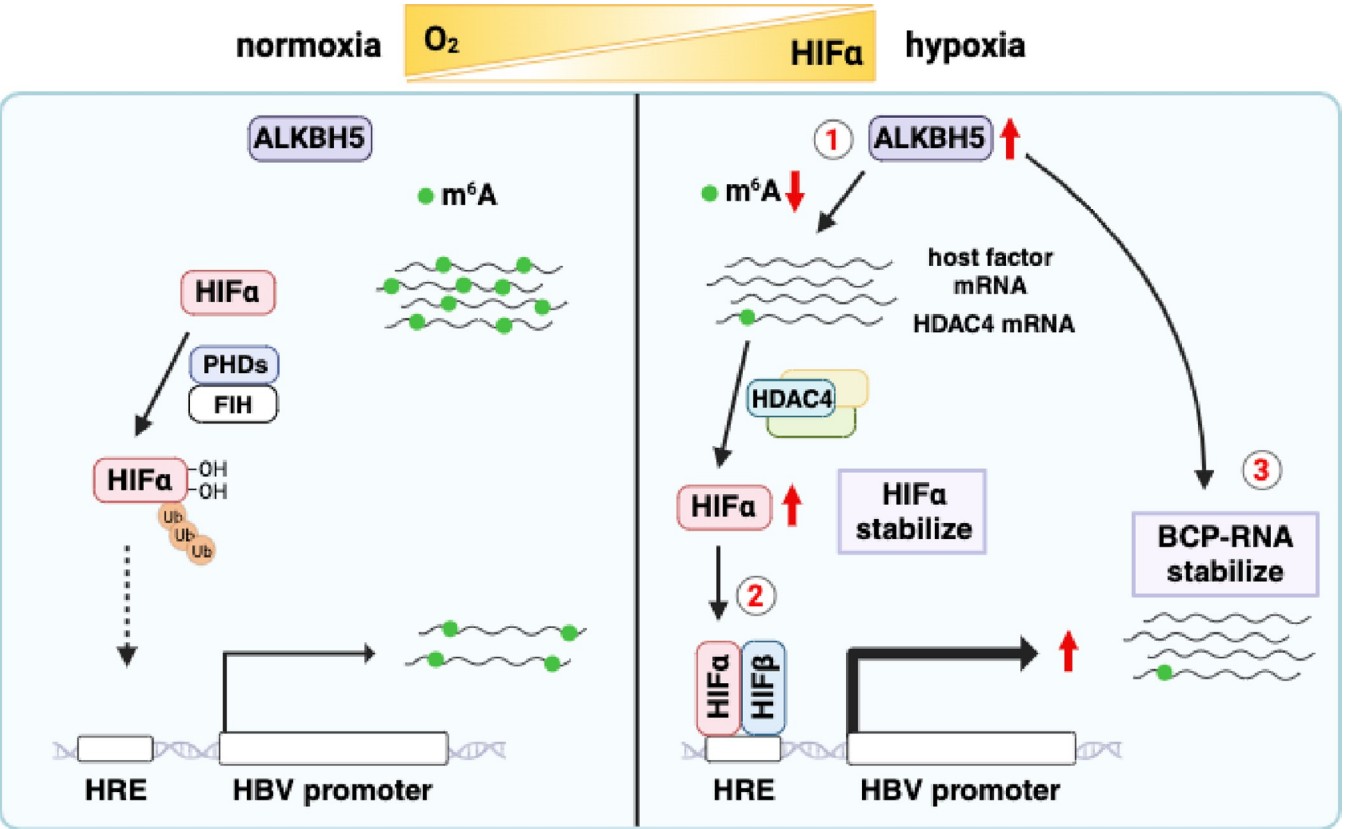

**Fig 7. A model for ALKBH5 regulation of HBV RNAs under hypoxic conditions. 1.** Hypoxia increases ALKBH5 protein expression and global reduction in m$^6$A methylated host and HBV BCP RNAs. **2.** Hypoxia increases HDAC4 expression and stabilizes HIF-1α and HIF-2α expression that can activate HBV transcription via binding to the basal core promoter. **3.** Reduced m$^6$A methylation increases the half-life of HBV BCP-RNAs.

PEG8000 for 16h. Viral inoculum was removed and cells washed three times with PBS. Infected cells were maintained in 5% $CO_2$ and 18% oxygen and treatments applied.

## Transfection

Plasmids were transfected into HepG2-NTCP cells using either polyethylenimine (PEI) or FuGENE HD Transfection Reagent (Promega) according to the manufacturer's protocol. siR-NAs for ALKBH5 (#1 s29686 and #2 s29688) and non-targeting control (4390844) were obtained from Thermo Fisher Scientific and transfected using DharmaFECT 4 Transfection Reagent (Horizon Discovery) according to the manufacturer's protocol.

## Quantitative PCR

Total cellular RNA was extracted using an RNeasy kit (Qiagen), then treated with TURBO DNA-free (Thermo Fisher Scientific) and reverse transcribed using a cDNA synthesis kit (PCR Biosystems) according to the manufacturer's protocol. Cellular DNA was extracted using QIAamp DNA kit (Qiagen). Gene expression was quantified using a SyGreen Blue Mix (PCR Biosystems) with the oligonucleotides listed in **S2 Table** using a qPCR program of 95˚C for 2 min followed by 45 cycles at 95˚C for 5 sec, 60˚C for 30 sec. Changes in gene expression were calculated by the ΔΔCt method relative to a housekeeper gene, β2-microglobulin (B2M).

## PacBio long read sequencing and analysis

HBV specific oligonucleotide enrichment and subsequent long-read sequencing and analysis was performed as reported [9]. Briefly, RNAs were prepared from normoxic or hypoxic HepG2-NTCP cells transfected with HBV HBV1.3 WT or m$^6$A-null constructs and 150ng reverse transcribed with barcoded sequencing primers. Oligonucleotide enrichment of HBV RNAs was performed as previously described [73]. Samples were sequenced using a Sequel II instrument to generate a PacBio 'Hifi library'. Reads were mapped to the HBV reference genome (genotype D3, ayw strain) using minimap2 [8,74]. HBV reads were assigned to previously reported TSS to identify canonical or unspliced transcripts [50] and splice junctions enumerated to identify non-canonical RNAs [8]. Incomplete sequences that did not encode the expected length of transcript and spliced RNAs that did not originate from the BCP region were not analysed. Differential gene expression was assessed using the Voom function in Limma (Bioconductor EdgeR script). The sequencing data is available via the SRA at NCBI (BioProject ID: PRJNA1000182).

## Fractionation of HBV BCP-RNAs in the nucleus, cytosol and core particles

HepG2-NTCP cells transfected with HBV1.3 plasmid were washed with PBS and lysed in 50 mM Tris-HCl, pH 8.0, and 1% NP-40 with protease inhibitor cocktail. After incubating cells at 4˚C for 20 min in the culture plate, the lysate was centrifuged for 5 min at 20,000 x g. The pellet was the nuclear fraction and RNA was extracted using Trizol. HBV core particles were isolated from the supernatant according to the protocol described by Belloni et al [75]. Briefly, 100 mM CaCl$_2$, DNase I, and RNase A were added to the supernatant and incubated for at 37˚C for 2 h. The supernatant was incubated in 5 mM EDTA, 7% PEG8000,1.75M NaCl, at 4˚C for 2 h. After centrifugation at 20,000 x g for 30 min at 4˚C, the supernatant was discarded and the capsid-containing pellet resuspended in TNE buffer (10 mM Tris-HCl (pH 8) 1mM EDTA, proteinase K), and RNA was extracted using Trizol. To calculate BCP-RNA levels in the cytosol, total HBV RNAs were extracted from the cells using Trizol and BCP-RNA levels in the nucleus and capsid subtracted after quantification by qPCR.

## Extracellular HBV DNA quantification

Extracellular HBV DNA was quantified according to the protocol described previously [76]. Briefly, culture supernatant was treated with DNase I (Thermo Fisher Scientific) at 37˚C for 60 minutes, then treated with 2x lysis buffer (100 mM Tris-HCl (pH7.4), 50 mM KCl, 0.25% Triton X-100, and 40% glycerol) containing 1mM EDTA. HBV DNA was amplified by qPCR using primers for HBV rcDNA and quantified against a DNA referent standard curve.

## Roadblock qPCR

Roadblock qPCR was performed according to the previously reported protocol [77]. HepG2-NTCP cells transfected with HBV1.3 plasmids and incubated under 18% or 1% oxygen conditions treated with 100 μM 4-thiouridine (4sU) for 0, 6, 12, or 24h 18% or 1% oxygen conditions. Total cellular RNA was extracted using an RNeasy (Qiagen) kit and a TURBO DNA-free Kit (Thermo Fisher Scientific), 1 μg of RNA was treated with 54mM N-ethylmaleimide (NEM) in NEM buffer [50mM Tris/HCl (pH 8.0) and 1mM EDTA] at 42˚C for 90 min. The reaction was stopped with 20 mM DTT, and RNA purified using a Zymo RNA Clean & Concentrator Kit. cDNA was synthesized from the purified RNA using SuperScript III First-Strand Synthesis System (Thermo Fisher Scientific) and HBV RNAs analyzed by qPCR.

## SDS-PAGE and western blot

Samples were lysed in RIPA buffer (50 mM Tris (pH 8.0), 150 mM NaCl, 1% Nonidet P-40, 0.5% sodium deoxycholate, and 0.1% sodium dodecyl sulphate) supplemented with protease inhibitor cocktail tablets (Roche). 4x Laemmli reducing buffer was added to samples before heating at 95˚C for 10 min. Proteins were separated on 8 or 14% polyacrylamide gel and transferred to activated 0.45 μm PVDF membranes (VWR, UK). Membranes were blocked in 5% skimmed milk and proteins detected using specific primary and HRP-conjugated secondary antibodies. Proteins were detected using a SuperSignal West Pico chemiluminescent substrate kit (Pierce) and images collected on a G:Box mini (Syngene).

## Immuno northern assay

10 μg of RNA was electrophoresed in a 1% MOPS agarose gel containing 2.2 M formaldehyde. 18 S and 28 S ribosomal RNA species were visualized under UV light after electrophoresis to verify the amount of RNA loaded and to assess degradation. After denaturation with 50 mM NaOH for 5 min, RNAs were transferred to a nylon membrane by capillary transfer using 20× SSC buffer. After UV crosslinking, the membrane was blocked in 5% skimmed milk and incubated with an anti-m$^6$A polyclonal antibody (Synaptic Systems) and HRP-conjugated secondary antibodies. Signals were realised using a SuperSignal West Pico chemiluminescent substrate kit (Pierce) and images collected on a G:Box mini (Syngene).

## MeRIP

Total cellular RNA was extracted using an RNeasy (Qiagen) kit and a TURBO DNA-free Kit (Thermo Fisher Scientific), the treated RNA was further purified using an RNeasy kit. 2 μg of RNA was incubated overnight at 4˚C with protein A agarose beads treated with Rabbit IgG or anti-m$^6$A polyclonal antibody (Synaptic Systems) in MeRIP buffer (Tris-HCl (pH 8.0),150 mM NaCl,0.1% Nonidet P-40, 1mM EDTA) supplemented with RNase inhibitor (Promega). Beads were washed 5 times with MeRIP buffer and bound RNA eluted with 6.7 mM m$^6$A Sodium salt. Eluted RNA was purified using a Qiagen RNA extraction kit and quantified by qPCR. Quantities of HBV BCP-RNA, *CREBBP*, *HDAC4* and *HIF-1α* were calculated relative to input total RNA.

## Supporting information

**S1 Fig. Hypoxic increase in host *CA9* gene expression, HBV BCP-RNAs and core antigen.** HepG2-NTCP cells were transfected with HBV plasmids (4h) and cultured at 18% or 1% $O_2$ for 72h. (**A**) Total cellular DNA was extracted at 4h post-transfection and HBV DNA measured, normalized to *PrP* housekeeping gene and expressed relative to HBV WT. Data are presented as mean ± S.D. of n = 3 from one of 3 independentexperiments. (**B-C**) *CA9* mRNA and HBV BCP-RNAs were quantified by qPCR, expressed relative to a *B2M* housekeeping gene and data presented as the mean ± S.D. of n = 9 samples from 3 independent experiments. Statistical significance was assessed using a non-parametric (Kruskall-Wallis) ANOVA, * $p < 0.05$, ** $p < 0.01$ *** $p < 0.001$, **** $p < 0.0001$. (**D**) HBc protein and β-Actin expression. (EPS)

**S2 Fig. Hypoxia perturbs TSS usage of HBV WT and m$^6$A-null spliced transcripts.** The number of pC or pg TSS spliced transcripts in HBV WT or m$^6$A-null infections at 18% or 1% oxygen were determined (S1 Table). Hypoxic conditions show a significant increase in the relative usage of the pgRNA TSS in WT but not m$^6$A-null samples (Student's t-test, p< 0.05). (EPS)

**S3 Fig. Hypoxic regulation of host gene expression of enzymes involved in m⁶A-modification pathway.** (**A**) Schematic of m⁶A-related host factors. (**B**) Gene expression of m⁶A related factors was analyzed by sequencing RNA from HepG2-NTCP cells incubated 18% or 1% oxygen for 72h (GSE ID: GSE186279). Log2 fold change in RNA levels of m⁶A related factors in hypoxic HepG2 cells. (**C**) Expression of methyltransferase complex and m⁶A-binding factors. RNA samples were extracted from HepG2-NTCP cells cultured under 18% or 1% oxygen for 72h and analyzed by qPCR. The data are expressed relative to *B2M* housekeeping gene and presented as mean ± S.D. of n = 4 samples from 2 independent experiments.
(EPS)

**S4 Fig. Efficiency of ALKBH5 siRNAs.** Efficiency of siRNA #1 and #2 targeting ALKBH5. HepG2-NTCP cells transfected with siRNA (NC; Non-targeting, ALK; ALKBH5) for 6h and cultured at 18% or 1% oxygen for 72h and RNA extracted. *ALKBH5* RNA and protein levels were detected by qPCR and western blotting, respectively. Statistical significance was assessed using Mann-Whitney tests, with Bonferroni correction for multiple comparisons, *** $p < 0.001$.
(EPS)

**S5 Fig. *HIF-1α* mRNA methylation.** Methylated RNA immunoprecipitation (MeRIP) assay. HepG2-NTCP cells were incubated in 18% or 1% oxygen for 72h. RNA was extracted and incubated overnight with protein A agarose beads coated with IgG or m⁶A antibody. The beads were washed 5 times, and bound RNA eluted with 6.7 mM m⁶A Sodium salt, purified using a Qiagen RNA extract kit and *HIF-1α* transcripts measured by qPCR. Data are expressed relative to input total RNA and represent the mean ± S.D. of n = 4 samples from 2 independent experiments.
(EPS)

**S6 Fig. Hydroxylated HIF-1α protein expression and *HDAC4* mRNA methylation in ALKBH5 silenced cells.** (**A**) HepG2-NTCP cells transfected with siRNA (siALKBH5-2) for 6h were cultured at 1% oxygen for 24h and treated with MG132 (20 μM) for 4h. Samples were collected and probed for HIF-1α, ALKBH5 and β-Actin by western blotting. (**B**) HepG2-NTCP cells were transfected siALKBH5 and incubated at 1% oxygen for 72h. RNA was extracted and incubated overnight with protein A agarose beads with IgG or m⁶A antibody. The beads were washed 5 times and bound RNA eluted in 6.7 mM m⁶A sodium salt, purified using a Qiagen RNA extract kit and *HDAC4* transcripts measured by qPCR. Data are expressed relative to input total RNA and represent the mean ± S.D. of n = 6 samples from 3 independent experiments.
(EPS)

**S1 Table. HBV transcript analysis by long-read sequencing.**
(XLSX)

**S2 Table. Oligonucleotide sequences used for qPCR.**
(XLSX)

## Acknowledgments

We thank Ulla Protzer (TUM, Germany) for providing HBV stocks, Stephan Urban (University of Heidelberg) for HepG2-NTCP cells, Azim Ansari for the enrichment probes and Esther Ng for advice in analysing and mapping HBV sequences.

## Author Contributions

**Conceptualization:** Senko Tsukuda.

**Formal analysis:** James M. Harris, Peter Balfe.

**Funding acquisition:** Jane A. McKeating.

**Investigation:** Senko Tsukuda.

**Supervision:** Jane A. McKeating.

**Writing – original draft:** Senko Tsukuda, Jane A. McKeating.

**Writing – review & editing:** James M. Harris, Andrea Magri, Peter Balfe, Aleem Siddiqui, Peter A.C. Wing.

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
