## [Decision Letter · Decision Letter 0]

9 Sep 2023

Dear Prof McKeating,

Thank you very much for submitting your manuscript "The N6-methyladenosine demethylase ALKBH5 regulates the hypoxic HBV transcriptome." for consideration at PLOS Pathogens. As with all papers reviewed by the journal, your manuscript was reviewed by members of the editorial board and by several independent reviewers. In light of the reviews (below this email), we would like to invite the resubmission of a significantly-revised version that takes into account the reviewers' comments.

The reviewers were excited about the potential impact of your work but identified several deficiencies in your study that would need to be addressed. Reviewer #1 asks for a more defined hypothesis and model, and reviewer #2 asks for HIF-1 / HIF-2 knockout controls. Of course, all reviewer comments would need to be addressed in any revised manuscript.

We cannot make any decision about publication until we have seen the revised manuscript and your response to the reviewers' comments. Your revised manuscript is also likely to be sent to reviewers for further evaluation.

Sincerely,

Robert F. Kalejta

Academic Editor

PLOS Pathogens

Blossom Damania

Section Editor

PLOS Pathogens

Kasturi Haldar

Editor-in-Chief

PLOS Pathogens

orcid.org/0000-0001-5065-158X

Michael Malim

Editor-in-Chief

PLOS Pathogens

orcid.org/0000-0002-7699-2064

The reviewers were excited about the potential impact of your work but identified several deficiencies in your study that would need to be addressed. Reviewer #1 asks for a more defined hypothesis and model, and reviewer #2 asks for HIF-1 / HIF-2 knockout controls. Of course, all reviewer comments would need to be addressed in any revised manuscript.

Reviewer's Responses to Questions

**Part I - Summary**

Reviewer #1: The manuscript by Tsukuda et al. addresses the contribution of the N6-methyladenosine (m6A) demethylase ALKBH5 to the hypoxia-mediated increase in HBV transcripts. The data in general are of good quality. However, some of the data and conclusions are confusing and sometimes appear to be inconsistent, and to some extent come across as a collection of potentially interesting datasets without a clearly established link/causality. This complicates the assessment if and how the data contribute to our understanding of HBV biology and the impact of hypoxia in this respect.

Reviewer #2: In this manuscript,the authors have investigated the mechanisms underpinning the regulation of the hepatitis B virus transcriptome in hypoxia. They conclude a role for N6-methyladenosine demethylase ALKBH5 in this response. This is a follow on from previous work from this group demonstrating a role for the hypoxia inducible factor (HIF) to activate the basal core promoter to transcribe pre-genomic RNA. Hypoxia increased viral RNA abundance in a manner associated with m6A methyladenosine modifications. The authors went on to investigate the impact of hypoxia on RNA demethylases and found increased expression of ALKBH5 in hepatocytes in a HIF dependent manner. Silencing ALKBH5 reduced HBV pre-genomic RNA. Chronic HBV infection remains a disease of global importance lacking in curative treatment and therefore research in this area is of importance. A key strength is the work on viral transcription. The authors, however need to address the questions I have outlined in relation to the role of HIF-1 and HIF-2 in this response.

**Part II – Major Issues: Key Experiments Required for Acceptance**

Reviewer #1: 1. A major comment is that I currently do not see the connection between different datasets and do not see what working hypothesis the authors have regarding their data. Based on the data, the way I see it: expression of the m6A demethylase ALKBH5 is increased during hypoxia, (ii) ALKBH5 then somehow (how exactly?) prevents degradation of HIF isoforms by the proteasome, leading (iii) to an increased HIF-mediated expression of viral mRNAs. But, in this scenario, why are the m6A sites in viral RNAs important (= Figure 1 and Figure 2 of the manuscript and earlier data) as these seem to have nothing to do with point (i) to (iii) above? The authors need to design additional assays to assess and explain potential causal relationships between their findings, e.g. how does ALKBH5 prevent degradation of HIF, how and where do the m6A sites in the viral transcripts come into play – currently these seem to be separate sets of observations. This should allow them to come up with a (possibly partly hypothetical) model of what they believe is going on.

2. Figure 3B: based on the Western blot, ALKBH6 protein levels seem to be only moderately increased in hypoxic conditions. Were independent replicates performed? Since this is a central point in the manuscript, the authors should quantify the apparent differences in protein levels and perform statistical analysis. This is especially important since in the Western blot shown in Supplementary Figure S3, there is no apparent increase in ALKBH5 protein levels in hypoxic (1% O2) versus ‘normal’ conditions (18% O2). The authors should explain and clarify these apparent conflicting results.

3. I do not agree with the statement (lines 193 and onwards) that ‘hypoxia did not alter the half-life of HBV WT pgRNA but we noted a significant reduction in m6A-null pgRNA under these conditions’. This refers to Figure 1E. In that figure, one can see that half-life of WT pgRNA goes down from 13.9h to 12.2h on average while the half-life of m6A-null pgRNA goes down from 8.3h to 6.9h on average. Hence, on average, half-life of WT pgRNA is reduced with 1.7h in hypoxic conditions, while that of m6Anull pgRNA is reduced with 1.4h in hypoxic conditions. That seems to be a very similar reduction. The standard deviation of this reduction is lower for m6Anull pgRNA compared to WT pgRNA, which likely explains the different outcome in statistical analysis. However, I very much doubt that this difference (if any) amounts to an actual biological difference. Hence, I do not agree with the statement. Additional assays would be needed to support such statement. For example, Roadblock qPCR assays could be performed to accurately assess whether there is a difference in mRNA degradation.

4. Lines 176-177, refering to Figure 1C: authors indicate that SP1 transcript levels were reduced in HBV m6Anull samples compared to WT, with a concomitant increase in SP14. This decrease/increase is not noticeable in Figure 1C – are these differences statistically significant?

5. Does hypoxia affect expression of any of the other relevant players in m6A biology (writer complex components, erasers, readers)?

Reviewer #2: 1) In the introduction and discussion, the authors should broaden their discussion of HIF to the well described and complex role of HIF in infection, inflammation and immunity.

2) In figures 1 and 2, the authors use CA9 mRNA as a surrogate for HIF activation. The authors need to also demonstrate the temporal effects of infection, hypoxia and hydroxylase inhibitor treatment on HIF-1 and HIF-2 levels in wild type and hepatocytes by western blot to confirm its activation and compare its activation levels under the various treatments applied.

3) The authors need to confirm that hypoxic activation of ALKBH5 expression is HIF-1 or HIF-2 dependent using HIF knock down in this system. Similarly, does knockdown of HIF-1 or HIF-2 (or both) reduce alterations in viral transcription in response to hypoxia this system.

**Part III – Minor Issues: Editorial and Data Presentation Modifications**

Reviewer #1: 1. Line 99: only METTL3 functions as a methyltransferase. METTL14 serves as a catalytically inactive allosteric activator of METTL3.

2. Line 100-101: suggest to rephrase to ‘This complex typically methylates adenosine residues within the consensus DRACH…’: methylation does not exclusively occur in this consensus sequence, so add ‘typically’.

3. Line 264: I presume the authors mean that beta-actin transcripts were unchanged (rather than HIF-1alpha)?

4. Line 279: I presume the authors mean that the data are ‘consistent with a role for the demethylase in preventing destabilization of both isoforms’ (rather than a role in destabilizing the isoforms as currently indicated)?

Reviewer #2: Minor:

1) Please avoid the use of the phrase “there was a trend albeit non-significant” (Page 8).

PLOS authors have the option to publish the peer review history of their article (what does this mean?). If published, this will include your full peer review and any attached files.

Reviewer #1: No

Reviewer #2: No

Figure Files:

Data Requirements:

Please note that, as a condition of publication, PLOS' data policy requires that you make available all data used to draw the conclusions outlined in your manuscript. Data must be deposited in an appropriate repository, included within the body of the manuscript, or uploaded as supporting information. This includes all numerical values that were used to generate graphs, histograms etc.. For an example see here on PLOS Biology: http://www.plosbiology.org/article/info:doi%2F10.1371%2Fjournal.pbio.1001908#s5.
---

## [Decision Letter · Decision Letter 1]

20 Dec 2023

Dear Prof McKeating,

We are pleased to inform you that your manuscript 'The N6-methyladenosine demethylase ALKBH5 regulates the hypoxic HBV transcriptome.' has been provisionally accepted for publication in PLOS Pathogens.

Best regards,

Robert F. Kalejta

Academic Editor

PLOS Pathogens

Blossom Damania

Section Editor

PLOS Pathogens

Kasturi Haldar

Editor-in-Chief

PLOS Pathogens

orcid.org/0000-0001-5065-158X

Michael Malim

Editor-in-Chief

PLOS Pathogens

orcid.org/0000-0002-7699-2064

Reviewer Comments (if any, and for reference):

Reviewer's Responses to Questions

**Part I - Summary**

Reviewer #1: I am pleased with the way how the authors have revised the manuscript according to the comments that were raised. I am also pleased by the notion that the suggestion to use Roadblock PCR has been picked up by the authors and has further strengthened some aspects of the manuscript.

Reviewer #2: The authors have addressed my concerns.

**Part II – Major Issues: Key Experiments Required for Acceptance**

Reviewer #1: (No Response)

Reviewer #2: The authors have addressed my concerns.

**Part III – Minor Issues: Editorial and Data Presentation Modifications**

Reviewer #1: (No Response)

Reviewer #2: The authors have addressed my concerns.

PLOS authors have the option to publish the peer review history of their article (what does this mean?). If published, this will include your full peer review and any attached files.

Reviewer #1: No

Reviewer #2: **Yes: **Prof. Cormac Taylor

---

## [Editor Report · Acceptance letter]

9 Jan 2024

Dear Prof McKeating,

We are delighted to inform you that your manuscript, "The N6-methyladenosine demethylase ALKBH5 regulates the hypoxic HBV transcriptome.," has been formally accepted for publication in PLOS Pathogens.

Best regards,

Michael Malim

Editor-in-Chief

PLOS Pathogens

orcid.org/0000-0002-7699-2064